# SP-LoRA: Sparsity-Preserved Low-Rank Adaptation for Sparse Large Language Model

## Abstract

Large Language Models (LLMs) excel in various natural language processing tasks but face significant hardware resource demands and inference latency due to their large parameter counts. To address these challenges, post-training pruning techniques like SparseGPT, Wanda, and RIA have been developed to reduce parameters. However, these methods often result in performance gaps, particularly for smaller models, and lack efficient fine-tuning strategies that preserve sparsity.

This paper presents SP-LoRA, a novel approach that integrates the advantages of low-rank adaptation (LoRA) with the efficiency of sparse models. Our method preserves sparsity when merging LoRA adapters with sparse matrices by introducing a mask matrix, $\mathcal{M}$. Additionally, to address the significant memory overhead associated with maintaining sparsity, we propose a hybrid technique that combines gradient checkpointing and memory reuse. This approach effectively reduces GPU memory usage during fine-tuning while achieving comparable efficiency to standard LoRA. Through extensive evaluations on sparse LLMs pruned by Wanda or SparseGPT, followed by fine-tuning with SP-LoRA, we demonstrate its effectiveness in both zero-shot scenarios and domain-specific tasks.

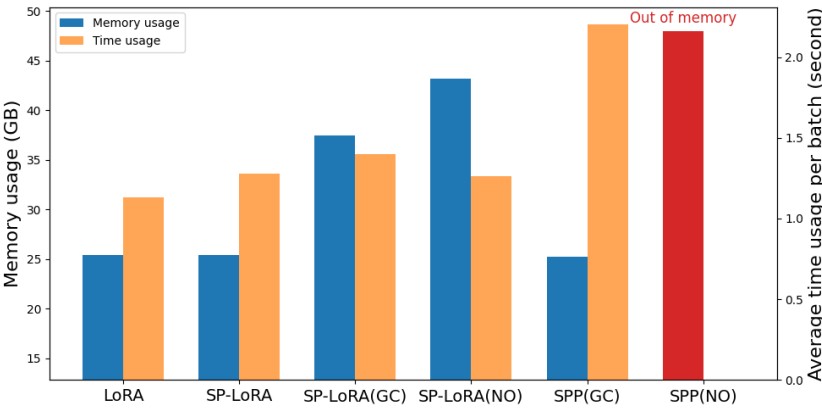

Figure 1: Memory and time usage of LoRA, SP-LoRA, and SPP, with GC denoting gradient checkpointing and NO representing no optimization (See Section 4.2 for details). Our approach SP-LoRA performs close to LoRA and outperforms the existing method SPP while preserving the sparsity.

## 1 Introduction

Large language models (LLMs) have exhibited exceptional performance across various natural language processing tasks, leading to their growing adoption. However, their extensive number of parameters demands substantial hardware resources for deployment, which limits accessibility. Additionally, the sheer scale of these models can slow down inference speed, posing challenges in applications where low latency is critical.

Various post-training unstructured pruning methods, such as SparseGPT (Frantar & Alistarh, 2023), Wanda (Sun et al., 2024), and RIA (Zhang et al., 2024), have been proposed to reduce model parameters and tackle the challenges mentioned earlier. These techniques require only a small number of samples and can transform a dense model into an unstructured or semi-structured sparse model in just a few minutes. While efficient and user-friendly, there remains a performance gap between the original dense model and the pruned sparse model, particularly for small- and medium-sized models under 2:4 semi-structured sparsity (Mishra et al., 2021). This gap hinders the practical application of these pruned sparse methods.

To utilize these models effectively, continuous pre-training is essential to compensate for the performance decline in sparse models. However, achieving desired performance through continuous pre-training can be quite costly. Therefore, there is an urgent need for efficient and low-resource tuning methods for sparse LLMs that preserve their sparsity. Unfortunately, current research has primarily concentrated on pruning strategies, with insufficient focus on the tuning of sparse models.

Contrasted with sparse language models, low-rank adaptation (LoRA; Hu et al., 2021) and other parameter-efficient fine-tuning (PEFT) techniques have been developed for dense language models to alleviate the computational burdens associated with various training phases. These methodologies facilitate the fine-tuning of dense LLMs with reduced resource requirements, thereby prompting the question: **Can LoRA be effectively utilized for the fine-tuning of sparse LLMs?**

In addressing this query, we introduce SP-LoRA, a simple yet effective method for preserving sparsity while performing low-rank adaptation on sparse LLMs. The primary challenge in applying LoRA to sparse LLMs lies in the fact that integrating LoRA's adapter with the weight matrix results in the loss of sparsity. To address this issue, we introduce an additional mask matrix $\mathcal{M}$, derived from the pruned weight matrix, as an extra weight term in LoRA. This mask delineates the locations of non-zero elements within the weight matrix $\mathcal{W}$, ensuring that sparsity is maintained throughout the training process. However, the introduction of this mask leads to an increased number of activations being tracked in the computational graph, consequently imposing a significantly higher GPU memory overhead for SP-LoRA compared to LoRA (See Section 3.2.1 for a detailed analysis). To address this issue, we propose a hybrid approach that combines gradient checkpointing (Chen et al., 2016) with memory reutilization techniques for SP-LoRA. This strategy minimizes unnecessary GPU memory allocation, making SP-LoRA as efficient as LoRA. Specifically, during each forward pass, we first compute the mask and generate the new weight matrix by merging the adapter, mask, and initial weight matrix. This process reuses the weight matrix to directly store the new weight matrix. In the backward pass, we recompute the mask, and then calculate the gradients of the input activations and adapters. Finally, we restore the initial weight matrix from the updated one for use in the next iteration's computation (see Section 3.2.2 for a detailed implementation).

We evaluate the proposed SP-LoRA on various LLMs. First, an LLM is pruned using a post-training pruning method, specifically Wanda or SparseGPT. Next, SP-LoRA is employed to fine-tune the pruned models using a portion of the collected pre-training and instruction data. We then directly assess the zero-shot performance of the tuned sparse LLM across a range of well-known text tasks. Additionally, we use SP-LoRA to fine-tune the sparse models on task-specific datasets, particularly for well-known challenging tasks, including math and code. This aims to explore the domain adaptation capabilities of SP-LoRA when addressing difficult problems.

The main contributions of this paper are summarized in the following:

(1) We propose SP-LoRA, a parameter-efficient fine-tuning method for sparse LLMs that preserves model sparsity during the fine-tuning process. This approach employs a hybrid technique that combines gradient checkpointing and memory reuse, effectively reducing the GPU memory overhead typically associated with fine-tuning sparse LLMs.

(2) Extensive experiments on sparse LLMs with various sparsity patterns and ratios demonstrate the effectiveness of SP-LoRA. As illustrated in Figure 1, SP-LoRA achieves comparable performance to LoRA—despite not preserving sparsity—in terms of memory and time usage. It significantly outperforms the sparsity-preserved SPP (Lu et al., 2024), especially regarding memory efficiency.

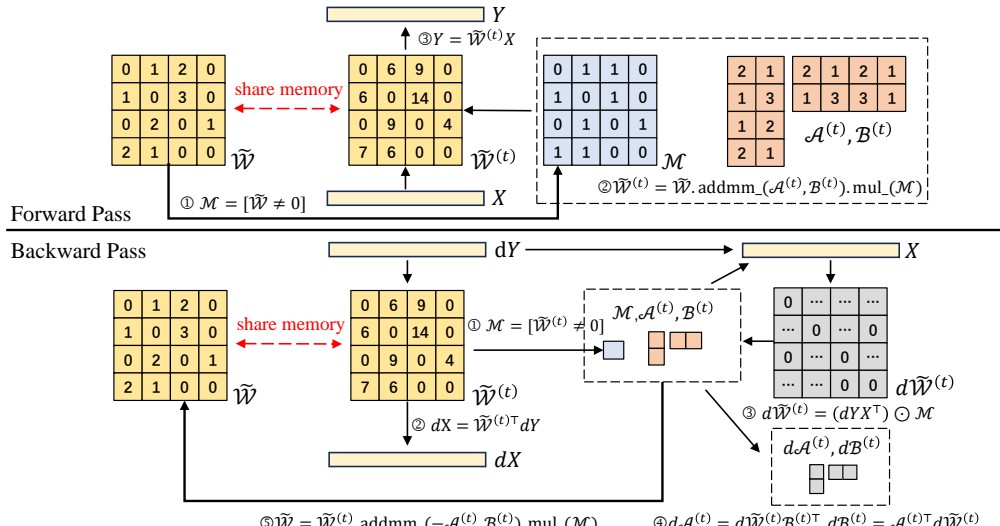

Figure 2: The workflow of SP-LoRA with memory optimization. We introduce an additional mask $\mathcal{M}$ into the LoRA framework to preserve the sparsity of the model. Meanwhile, the memory overhead of SP-LoRA is optimized by reutilizing the memory of $\tilde{\mathcal{W}}$ to store weight matrix $\tilde{\mathcal{W}}^{(t)}$ and by recomputing the mask $\mathcal{M}$.

## 2 RELATED WORK

### 2.1 PRUNING

Pruning (Han et al., 2016) is a promising technique for compressing neural networks by removing unimportant weights. From the perspective of sparse structure, pruning methods can be categorized into structured (Ashkboos et al., 2024; Chen et al., 2024; Hu et al., 2024; Liu et al., 2024; Men et al., 2024; Muralidharan et al., 2024) and unstructured pruning (Frantar & Alistarh, 2023; Sun et al., 2024; Zhang et al., 2024). Structured pruning achieves compression by selectively eliminating entire structural units such as channels, filters, attention heads, or layers from the neural network. Conversely, unstructured pruning achieves compression by removing individual unimportant elements from the weight matrices, effectively transforming dense matrices into sparse ones. And thanks to hardware developments, models obtained with unstructured pruning can also be efficiently accelerated when using a specific sparse structure, such as 2:4 sparsity (Mishra et al., 2021).

From the perspective of optimization methods, pruning techniques can be further classified into training-based and post-training pruning. Training-based pruning (Louizos et al., 2018; Sanh et al., 2020) progressively thins out a dense model during the training phase. This approach typically involves introducing masks into the model and controlling its sparsity through an additional regularization loss computed based on these masks. Although widely applicable to smaller models, training-based pruning is challenging to implement for larger models due to the substantial increase in GPU memory overhead and the requirement for extensive training data. Consequently, there has been a growing interest in post-training methods (Frantar & Alistarh, 2023; Sun et al., 2024; Zhang et al., 2024) that enable pruning with a small number of calibration data, particularly for large LLMs.

### 2.2 PARAMETER-EFFICIENT FINE-TUNING

PEFT methods are designed to fine-tune pre-trained models with minimal trainable parameters. Typically these methods freeze the original model and insert a series of trainable adapters, including but not limited to prefix tokens (Liu et al., 2022), side networks (Zhang et al., 2020), parallel and serial adapters (Houlsby et al., 2019; Hu et al., 2023). These techniques are particularly advantageous when working with large pre-trained models, as full fine-tuning of all parameters can be both computationally prohibitive and data-intensive. Among these methods, LoRA and its variants (Hu et al.,

2021; Zhang et al., 2023; Zhao et al., 2024) are the most widely adopted PEFT approaches, offering the benefit of merging the adapter's parameters with the model weights post-training. However, for sparse LLMs, this merging process can transform the sparse model into a dense one, thereby undermining the benefits of sparsity. In this work, we aim to enhance LoRA to make it compatible with sparse LLMs.

### 2.3 SPARSITY PRESERVED TRAINING

Contrary to pruning, which transforms a dense model into a sparse one, some approaches aim to train a sparse model from scratch or an existing sparse model. We refer to these techniques as sparsity-preserved training methods, which include STE (Zhou et al., 2021), RigL (Evci et al., 2021), and others (Huang et al., 2024; Kurtic et al., 2023). These methods can produce sparse models that perform comparably to dense models; however, they require the training of all the parameters of the model and even require more GPU memory than the training of dense models, thereby posing challenges for application to LLMs. Recent work SPP (Lu et al., 2024), has proposed to reduce the training cost of sparse models by combining PEFT methods with sparsity-preserved training. SPP can be viewed as a variant of LoRA, using a special form of matrices as adapters and introducing additional weight terms in LoRA. SPP in the forward pass requires the construction of a matrix with the same size as the weight matrix and recording it in the computational graph. Therefore, despite requiring only a limited number of trainable parameters, SPP still encounters the issue of high GPU memory overhead. This work will address the high GPU overhead issue for sparsity-preserved training.

## 3 METHOD

In this section, we first review unstructured pruning and low-rank adaptation (Section 3.1), then introduce our proposed method, SP-LoRA (Section 3.2). We subsequently discuss the challenges of training sparse LLMs while preserving sparsity (Section 3.2.1) and explain how our approach addresses these challenges (Section 3.2.2).

### 3.1 PRELIMINARY

**Unstructured Pruning** Unstructured pruning methods are employed to transform the dense weight matrices of LLMs into sparse matrices. Let $\mathcal{W}$ denote a weight matrix of an LLM. The objective of unstructured pruning is to determine a mask $\mathcal{M}$ and weight updates $\Delta\mathcal{W}$, such that the dense matrix can be transformed into a sparse matrix $\tilde{\mathcal{W}}$. Mathematically, this transformation is expressed as: $\tilde{\mathcal{W}} = \mathcal{M} \odot (\mathcal{W} + \Delta\mathcal{W})$, where $\mathcal{W} \in \mathbb{R}^{R \times C}$, $\mathcal{M} \in \{0, 1\}^{R \times C}$, and $\Delta\mathcal{W} \in \mathbb{R}^{R \times C}$. $R$ and $C$ represent the number of rows and columns of the weight matrix, respectively.

**LoRA** LoRA is a method for adapting LLMs to specific tasks or domains by training only a small number of parameters. Its mathematical formulation is given by: $\mathcal{W}^{(t)} = \mathcal{W} + \mathcal{A}^{(t)} \times \mathcal{B}^{(t)}$, where $\mathcal{W}$ denotes the initial weight matrix, $\mathcal{W}^{(t)}$ represents the weight matrix at the $t$-th iteration of training, and $\mathcal{A}$ and $\mathcal{B}$ are the introduced trainable adapters, $\mathcal{A}^{(t)}$ and $\mathcal{B}^{(t)}$ represent the adapters at the $t$-th iteration of training. Here, $\mathcal{W} \in \mathbb{R}^{R \times C}$, $\mathcal{A} \in \mathbb{R}^{R \times r}$, $\mathcal{B} \in \mathbb{R}^{r \times C}$, and $r$ is much smaller than $R$ and $C$. During training, all parameters except $\mathcal{A}$ and $\mathcal{B}$ remain frozen.

### 3.2 SP-LoRA

To preserve the sparsity of the model, we adopt a simple approach by introducing a mask as an additional weighting term in the LoRA framework. Let us consider a sparse LLM with a weight matrix $\tilde{\mathcal{W}}$ and its corresponding mask $\mathcal{M}$. Based on LoRA, we first introduce adapters $\mathcal{A}$ and $\mathcal{B}$ for the weight matrix $\tilde{\mathcal{W}}$. Then, we incorporate the mask to ensure the sparsity of the weight matrix at each training iteration $t$:

$$\tilde{\mathcal{W}}^{(t)} = \tilde{\mathcal{W}} + \mathcal{M} \odot (\mathcal{A}^{(t)} \times \mathcal{B}^{(t)}). \tag{1}$$

We refer to this LoRA variant as SP-LoRA, which stands for Sparsity Preserved Low-Rank Adaptation. However, the introduction of the mask while ensuring the sparsity of the weights, alters the computational graph of LoRA, thus incurring significant GPU memory overhead, posing practical challenges for its implementation. Consequently, we will first analyze the cause of this high GPU memory overhead and propose a solution to address this issue.

### 3.2.1 MEMORY COMPLEXITY

Assuming that the current iteration is the $t$-th training step, let the input to the weight matrix be denoted as $X \in \mathbb{R}^{C \times L}$. For LoRA, the output can be represented as

$$Y = \tilde{\mathcal{W}}X + \mathcal{A}^{(t)}\mathcal{B}^{(t)}X. \tag{2}$$

This formulation corresponds to the following computational steps:

$$I_a^1 = \tilde{\mathcal{W}}X, \quad I_a^2 = \mathcal{B}^{(t)}X, \quad I_a^3 = \mathcal{A}^{(t)}I_a^2, \quad Y = I_a^1 + I_a^3, \tag{3}$$

where $I_a^1 \in \mathbb{R}^{R \times L}$, $I_a^2 \in \mathbb{R}^{r \times L}$, and $I_a^3 \in \mathbb{R}^{R \times L}$ represent the intermediate activations. In the context of back-propagation, the gradients for the parameters $\mathcal{A}^{(t)}$, $\mathcal{B}^{(t)}$, and $X$ must be computed. Given the gradient of $Y$ as $dY$, the gradients can be formulated as follows:

$$d\mathcal{A}^{(t)} = dY I_a^{2\top}, \quad dI_a^2 = \mathcal{A}^{(t)\top}dY, \quad d\mathcal{B}^{(t)} = dI_a^2 X^\top, \quad dX = \tilde{\mathcal{W}}^\top dY + \mathcal{B}^{(t)\top}dI_a^2. \tag{4}$$

Consequently, during the forward pass, GPU memory must be allocated for the intermediate activations $I_a^1$, $I_a^2$, and $I_a^3$, along with the output activation $Y$, encompassing a total of $rL + 3RL$ parameters. Additionally, the input activation $X$ and the intermediate activation $I_a^2$ are retained for back-propagation, involving $rL + CL$ parameters. During the backward pass, GPU memory allocation is required for the gradients $d\mathcal{A}^{(t)}$, $dI_a^2$, $d\mathcal{B}^{(t)}$, and $dX$, totaling $rR + rL + rC + CL$ parameters.

Then, considering the proposed method SP-LoRA, the mathematical expression for the output can be written as

$$Y = \{\tilde{\mathcal{W}} + \mathcal{M} \odot (\mathcal{A}^{(t)} \times \mathcal{B}^{(t)})\}X. \tag{5}$$

Compared to LoRA, which first multiply $X$ with $\mathcal{B}^{(t)}$ and then with $\mathcal{A}^{(t)}$, SP-LoRA needs to compute $\mathcal{M} \odot (\mathcal{A}^{(t)} \times \mathcal{B}^{(t)})$ first, corresponding to the following computational steps:

$$I_w^1 = \mathcal{A}^{(t)}\mathcal{B}^{(t)}, \quad \mathcal{M} = [\tilde{\mathcal{W}} \neq 0], \quad I_w^2 = \mathcal{M} \odot I_w^1, \quad I_w^3 = \tilde{\mathcal{W}} + I_w^2, \quad Y = I_w^3 X, \tag{6}$$

where $I_w^1, I_w^2, I_w^3 \in \mathbb{R}^{R \times C}$ represent the intermediate weights. The corresponding back-propagation process is outlined as follows:

$$dI_w^3 = dY X^\top, \ dX = I_w^{3\top}dY, \ dI_w^1 = dI_w^3 \odot \mathcal{M}, \ d\mathcal{A}^{(t)} = dI_w^1 \mathcal{B}^{(t)\top}, \ d\mathcal{B}^{(t)} = \mathcal{A}^{(t)\top}dI_w^1. \tag{7}$$

Hence, for SP-LoRA , during the forward pass, GPU memory allocation is necessary for the intermediate weights $\mathcal{M}$, $I_w^1$, $I_w^2$, $I_w^3$, and the output activation $Y$, encompassing a total of $4RC + RL$ parameters ($> rL + 3RL$). Additionally, the input activation $X$, the intermediate weights $\mathcal{M}$, and $I_w^3$ must be retained for the back-propagation process, involving $2RC + CL$ parameters ($> rL + CL$). In the backward pass, GPU memory must be allocated for the gradients $dI_w^1$, $dI_w^3$, $dX$, $d\mathcal{A}^{(t)}$, and $d\mathcal{B}^{(t)}$, summing to $2RC + CL + rR + rC$ parameters ($> rR + rL + rC + CL$).

Comparing the number of parameters retained for back-propagation by SP-LoRA and LoRA, it becomes evident that including masks significantly increases GPU memory overhead, despite not increasing the number of trainable parameters. In addition, SP-LoRA also allocates more temporary GPU memory than LoRA for both forward and backward, thus increasing the time overhead. Consequently, optimizing the GPU memory usage of SP-LoRA is imperative.

---

**Algorithm 1:** SP-LoRA Forward Pass

---

**Input:** Activation $X$, Sparse weight matrix $\tilde{\mathcal{W}}$, SP-LoRA adapters $\mathcal{A}^{(t)}, \mathcal{B}^{(t)}$.
**Output:** Activation $Y$

1 Compute mask: $\mathcal{M} = [\tilde{\mathcal{W}} \neq 0]$;
2 Update $\tilde{\mathcal{W}}$ to $\tilde{\mathcal{W}}^{(t)}$ in-place: $\tilde{\mathcal{W}}^{(t)} = \tilde{\mathcal{W}}.\text{addmm\_}(\mathcal{A}^{(t)}, \mathcal{B}^{(t)}).\text{mul\_}(\mathcal{M})$;
3 Save $X$ into context for backward;
4 Compute $Y$: $Y = \tilde{\mathcal{W}}^{(t)} X$;

---

**Algorithm 2:** SP-LoRA Backward Pass

---

**Input:** Gradient $dY$, Activation $X$, Sparse weight matrix $\tilde{\mathcal{W}}^{(t)}$, SP-LoRA adapters $\mathcal{A}^{(t)}, \mathcal{B}^{(t)}$.
**Output:** Gradients $d\mathcal{A}^{(t)}, d\mathcal{B}^{(t)}$, and $dX$

1 Compute mask: $\mathcal{M} = [\tilde{\mathcal{W}}^{(t)} \neq 0]$;
2 Compute gradient of $X$: $dX = \tilde{\mathcal{W}}^{(t)\top} dY$;
3 Compute gradient of $\tilde{\mathcal{W}}^{(t)}$: $d\tilde{\mathcal{W}}^{(t)} = (dY X^\top).\text{mul\_}(\mathcal{M})$;
4 Compute gradient of $\mathcal{A}^{(t)}$: $d\mathcal{A}^{(t)} = d\tilde{\mathcal{W}}^{(t)} \mathcal{B}^{(t)\top}$;
5 Compute gradient of $\mathcal{B}^{(t)}$: $d\mathcal{B}^{(t)} = \mathcal{A}^{(t)\top} d\tilde{\mathcal{W}}^{(t)}$;
6 Update $\tilde{\mathcal{W}}^{(t)}$ to $\tilde{\mathcal{W}}$ in-place: $\tilde{\mathcal{W}} = \tilde{\mathcal{W}}^{(t)}.\text{addmm\_}(-\mathcal{A}^{(t)}, \mathcal{B}^{(t)}).\text{mul\_}(\mathcal{M})$;

---

### 3.2.2 MEMORY OPTIMIZATION

We propose a hybrid gradient checkpointing and memory reutilizing approach to optimize memory usage. During the forward propagation phase of SP-LoRA, memory allocation is required for intermediate weights denoted as $\mathcal{M}, I_w^1, I_w^2$, and $I_w^3$. Despite their substantial demand on GPU memory, these intermediate weights entail minimal computational effort. Therefore, instead of providing extra memory for storing these intermediate weights, we can either recompute them during back-propagation or reuse existing memory to store them. Algorithm 1 and 2 provide the pseudo-code[1] detailing the forward and backward passes of SP-LoRA, respectively. Specifically, in the forward pass, we compute the weight matrix $\tilde{\mathcal{W}}^{(t)}$ and leverage the existing memory footprint of $\tilde{\mathcal{W}}$ to store it (Algorithm 1 Line 2). Upon transitioning to the backward propagation phase, we first recompute the mask $\mathcal{M}$ (Algorithm 2 Line 1), then the gradients of the weight matrices $\mathcal{A}^{(t)}$ and $\mathcal{B}^{(t)}$, alongside the input activation $X$, are computed (Algorithm 2 Line 2, 3, 4 and 5). Subsequently, we restore $\tilde{\mathcal{W}}$ from $\tilde{\mathcal{W}}^{(t)}$ (Algorithm 2 Line 6). The operational workflow of the optimized SP-LoRA is illustrated in Figure 2.

Refer to the Formula 6 and 7, after memory optimization, the requisite GPU memory allocation is confined to the parameters $\mathcal{M}$ and $Y$, encompassing $RC + RL$ parameters (a reduction from the initial $4RC + RL$). Similarly, only the input activation $X$, comprising $CL$ parameters (a decrease from the original $2RC + CL$), needs to be retained for the back-propagation process. During the backward pass, memory allocation is necessary for the gradients $dX, d\tilde{\mathcal{W}}^{(t)}, d\mathcal{A}^{(t)}$, and $d\mathcal{B}^{(t)}$, along with the mask $\mathcal{M}$, totaling $2RC + CL + rR + rC$ parameters, consistent with the memory requirements before optimization.

While this optimization incurs an additional computational cost of $rR + rC + 2RC$ FLOPs (Algorithm 2 Line 6), this increment is relatively insignificant against the total computational FLOPs ($\approx RCL$). As shown in Figure 1, the optimized SP-LoRA achieves similar time and memory overheads with LoRA, thereby ensuring its practical viability.

| Model | Mehtod | Sparsity | ARC-c | ARC-e | BoolQ | Hellaswag | OBQA | RTE | Winogrande | Average |
|---|---|---|---|---|---|---|---|---|---|---|
| Llama-2-7B | None | None | 43.52 | 76.35 | 77.74 | 57.14 | 31.40 | 62.82 | 69.06 | 59.72 |
| | SparseGPT | 2:4 | 31.31 | 63.93 | 68.90 | 43.54 | 24.60 | 63.18 | 65.90 | 51.62 |
| | SparseGPT+SPP | 2:4 | 34.30 | 67.38 | 68.29 | 50.54 | 27.00 | 64.26 | 66.93 | 54.10 |
| | SparseGPT+LoRA | None | 35.58 | 68.86 | 66.76 | 50.92 | 27.00 | 66.79 | 66.61 | 54.65 |
| | SparseGPT+SP-LoRA | 2:4 | 34.98 | 68.27 | 66.61 | 50.79 | 27.00 | 63.18 | 66.77 | 53.94 |
| | Wanda | 2:4 | 30.03 | 61.95 | 68.32 | 41.21 | 24.20 | 53.07 | 62.35 | 48.73 |
| | Wanda+SPP | 2:4 | 34.81 | 68.39 | 70.03 | 49.56 | 26.60 | 57.40 | 65.43 | 53.17 |
| | Wanda+LoRA | 2:4 | 36.01 | 69.19 | 71.71 | 50.61 | 27.00 | 58.84 | 64.72 | 54.01 |
| | Wanda+SP-LoRA | 2:4 | 35.75 | 70.29 | 70.43 | 50.33 | 27.60 | 60.29 | 64.48 | 54.16 |
| Llama-2-13B | None | None | 48.38 | 79.42 | 80.55 | 60.04 | 35.20 | 65.34 | 72.30 | 63.03 |
| | SparseGPT | 2:4 | 37.29 | 69.07 | 79.05 | 48.00 | 25.80 | 58.84 | 69.14 | 55.31 |
| | SparseGPT+SPP | 2:4 | 40.78 | 72.43 | 76.82 | 55.23 | 29.20 | 59.21 | 68.75 | 57.49 |
| | SparseGPT+LoRA | None | 39.76 | 72.81 | 76.54 | 55.51 | 31.20 | 66.79 | 69.61 | 58.89 |
| | SparseGPT+SP-LoRA | 2:4 | 39.85 | 72.90 | 76.30 | 55.65 | 30.00 | 67.51 | 69.38 | 58.80 |
| | Wanda | 2:4 | 34.47 | 68.48 | 75.72 | 46.39 | 24.40 | 57.04 | 66.69 | 53.31 |
| | Wanda+SPP | 2:4 | 40.02 | 71.51 | 75.72 | 54.55 | 29.40 | 62.09 | 69.61 | 55.56 |
| | Wanda+LoRA | None | 41.38 | 72.35 | 76.24 | 55.12 | 29.60 | 63.18 | 68.75 | 58.09 |
| | Wanda+SP-LoRA | 2:4 | 40.44 | 72.39 | 75.66 | 55.05 | 30.40 | 59.93 | 67.56 | 57.35 |
| Llama-3-8B | None | None | 50.26 | 80.09 | 81.35 | 60.18 | 34.80 | 69.31 | 72.38 | 64.05 |
| | SparseGPT | 2:4 | 32.00 | 62.67 | 73.70 | 43.19 | 22.20 | 53.79 | 65.75 | 50.47 |
| | SparseGPT+SPP | 2:4 | 39.42 | 69.95 | 71.93 | 51.67 | 25.80 | 63.18 | 68.27 | 55.75 |
| | SparseGPT+LoRA | None | 38.74 | 70.03 | 75.54 | 52.24 | 28.80 | 59.93 | 67.01 | 56.04 |
| | SparseGPT+SP-LoRA | 2:4 | 38.14 | 70.29 | 75.87 | 52.35 | 26.80 | 63.90 | 67.56 | 56.42 |
| | Wanda | 2:4 | 26.45 | 55.93 | 66.18 | 37.51 | 18.60 | 52.71 | 60.06 | 45.35 |
| | Wanda+SPP | 2:4 | 36.77 | 67.39 | 72.97 | 49.49 | 25.80 | 59.21 | 64.88 | 53.79 |
| | Wanda+LoRA | None | 37.12 | 69.11 | 73.61 | 50.94 | 27.60 | 59.21 | 66.38 | 54.85 |
| | Wanda+SP-LoRA | 2:4 | 38.31 | 69.53 | 71.56 | 50.83 | 28.00 | 54.87 | 66.30 | 54.20 |

Table 1: Zero-shot evaluation results of 7 tasks from EleutherAI LM Harness with models trained on a subset of the SlimPajama dataset with 0.5B tokens.

| | SparseGPT | | | Wanda | | |
|---|---|---|---|---|---|---|
| | SPP | LoRA | SP-LoRA | SPP | LoRA | SP-LoRA |
| SlimPajama-0.5B | 7.33 | 7.09 | 7.10 | 7.39 | 7.12 | 7.13 |
| Stanford Alpaca | 8.19 | 9.73 | 9.34 | 8.42 | 9.83 | 10.16 |

Table 2: Perplexity of pruned Llama-2-7B on wikitext2 after fine-tuning through SlimPajama-0.5B and Alpaca datasets respectively.

## 4 EXPERIMENTS

In this section, we will illustrate the effectiveness of SP-LoRA in training sparse LLMs through experiments.

**Experiment Setup**  We conducted our experiments using the Llama-2-7B, Llama-2-13B, Llama-3-8B and Llama-3.1-8B-instruct models (Touvron et al., 2023a;b; Dubey et al., 2024). Initially, we applied post-training pruning techniques, specifically SparseGPT and Wanda, with the **2:4 sparsity** type. Subsequently, the pruned models were fine-tuned using three distinct datasets: pre-training, instruction, and domain-specific. During fine-tuning, adapters were added to all sparse weight matrices within the model.

---

[1]addmm_ and mul_ are APIs in PyTorch for implementing in-place matrix multiplication and element-wise multiplication.

| Model | Mehtod | Sparsity | ARC-c | ARC-e | BoolQ | Hellaswag | OBQA | RTE | Winogrande | Average |
|-------|--------|----------|-------|-------|-------|-----------|------|-----|------------|---------|
| Llama-2-7B | None | None | 43.52 | 76.35 | 77.74 | 57.14 | 31.40 | 62.82 | 69.06 | 59.72 |
| | SparseGPT | 2:4 | 31.31 | 63.93 | 68.90 | 43.54 | 24.60 | 63.18 | 65.90 | 51.62 |
| | SparseGPT+SPP | 2:4 | 36.86 | 69.15 | 72.91 | 50.67 | 28.80 | 62.45 | 66.30 | 55.31 |
| | SparseGPT+LoRA | None | 35.67 | 63.13 | 70.73 | 51.19 | 26.40 | 70.40 | 64.09 | 54.52 |
| | SparseGPT+SP-LoRA | 2:4 | 36.01 | 64.35 | 72.17 | 51.84 | 29.60 | 59.93 | 63.61 | 53.93 |
| | Wanda | 2:4 | 30.03 | 61.95 | 68.32 | 41.21 | 24.20 | 53.07 | 62.35 | 48.73 |
| | Wanda+SPP | 2:4 | 36.26 | 69.44 | 72.02 | 49.64 | 27.80 | 55.96 | 63.77 | 53.56 |
| | Wanda+LoRA | None | 35.32 | 64.18 | 71.99 | 50.60 | 28.40 | 60.65 | 63.14 | 53.47 |
| | Wanda+SP-LoRA | 2:4 | 35.41 | 65.03 | 72.39 | 50.18 | 30.00 | 60.29 | 62.67 | 53.71 |
| Llama-2-13B | None | 2:4 | 48.38 | 79.42 | 80.55 | 60.04 | 35.20 | 65.34 | 72.30 | 63.03 |
| | SparseGPT | 2:4 | 37.29 | 69.07 | 79.05 | 48.00 | 25.80 | 58.84 | 69.14 | 55.31 |
| | SparseGPT+SPP | 2:4 | 42.06 | 73.32 | 78.62 | 55.02 | 29.40 | 65.70 | 69.77 | 59.13 |
| | SparseGPT+LoRA | None | 40.78 | 67.93 | 76.48 | 54.68 | 29.40 | 71.12 | 69.38 | 58.54 |
| | SparseGPT+SP-LoRA | 2:4 | 43.00 | 70.37 | 76.88 | 55.91 | 31.60 | 68.95 | 70.17 | 59.55 |
| | Wanda | 2:4 | 34.47 | 68.48 | 75.72 | 46.39 | 24.40 | 57.04 | 66.69 | 53.31 |
| | Wanda+SPP | 2:4 | 41.89 | 72.73 | 77.37 | 54.84 | 30.40 | 65.34 | 68.27 | 58.69 |
| | Wanda+LoRA | None | 40.02 | 68.35 | 76.09 | 54.17 | 29.80 | 64.98 | 66.93 | 57.19 |
| | Wanda+SP-LoRA | 2:4 | 39.42 | 69.40 | 78.01 | 55.16 | 30.00 | 72.20 | 67.80 | 58.86 |
| Llama-3-8B | None | 2:4 | 50.26 | 80.09 | 81.35 | 60.18 | 34.80 | 69.31 | 72.38 | 64.05 |
| | SparseGPT | 2:4 | 32.00 | 62.67 | 73.70 | 43.19 | 22.20 | 53.79 | 65.75 | 50.47 |
| | SparseGPT+SPP | 2:4 | 40.78 | 71.09 | 75.35 | 52.01 | 26.40 | 59.93 | 67.88 | 56.21 |
| | SparseGPT+LoRA | 2:4 | 38.31 | 65.45 | 76.79 | 50.51 | 28.20 | 54.51 | 62.98 | 53.82 |
| | SparseGPT+SP-LoRA | 2:4 | 38.05 | 64.02 | 73.27 | 48.89 | 25.20 | 60.65 | 62.12 | 53.17 |
| | Wanda | 2:4 | 26.45 | 55.93 | 66.18 | 37.51 | 18.60 | 52.71 | 60.06 | 45.35 |
| | Wanda+SPP | 2:4 | 38.48 | 68.64 | 74.77 | 49.53 | 25.20 | 58.48 | 64.64 | 54.25 |
| | Wanda+LoRA | 2:4 | 38.05 | 64.02 | 73.27 | 48.89 | 25.20 | 60.65 | 62.12 | 53.17 |
| | Wanda+SP-LoRA | 2:4 | 37.46 | 65.07 | 73.36 | 49.48 | 26.00 | 63.18 | 62.75 | 53.90 |

Table 3: Zero-shot evaluation results of 7 tasks from EleutherAI LM Harness with models trained on the Alpaca dataset.

| Model | Sparsity | ARC-c | ARC-e | BoolQ | Hellaswag | OBQA | RTE | Winogrande | Average |
|-------|----------|-------|-------|-------|-----------|------|-----|------------|---------|
| Llama-3.1-8B-instruct | None | 51.71 | 81.86 | 84.07 | 59.10 | 33.80 | 67.87 | 73.95 | 64.62 |
| +SparseGPT | 2:4 | 34.30 | 65.45 | 77.74 | 43.56 | 22.20 | 61.73 | 66.30 | 53.04 |
| +SP-LoRA | | | | | | | | | |
|   +FineWeb-Edu-5B | 2:4 | 43.60 | 77.90 | 76.36 | 54.19 | 32.40 | 64.62 | 69.85 | 59.85 |
|   +FineWeb-Edu-5B & Alpaca | 2:4 | 44.80 | 74.54 | 77.98 | 55.86 | 34.80 | 67.87 | 70.01 | 60.83 |

Table 4: Zero-shot evaluation results of 7 tasks from EleutherAI LM Harness with Llama-3.1-8B-instruct model trained on the FineWeb-edu-5B and Alpaca dataset.

- For the pre-training data, we utilized a subset of the SlimPajama dataset (Penedo et al., 2023), consisting of 0.5B tokens. After continual pre-train the model, we tested the model's zero-shot performance on seven datasets selected from EleutherAI LM Harness (Gao et al., 2024), including ARC-c, ARC-e (Clark et al., 2018), BoolQ (Clark et al., 2019), Hellaswag (Zellers et al., 2019), OBQA (Mihaylov et al., 2018), RTE, and Winogrande (Sakaguchi et al., 2019). During the training, the rank of adapters is set to 16, the batch size is set to 256k tokens, and the learning rate is set to $1 \times 10^{-3}$.

- For the instruction data, we use the Stanford-Alpaca dataset (Taori et al., 2023). After fine-tuning the model, we tested the model's zero-shot performance as above. During the training, the rank of adapters is set to 16, the batch size is set to 32 samples, and the learning rate is set to $1 \times 10^{-3}$.

- For the domain-specific dataset, we consider three domains: chat, math, and code. Specially, we used a 52k subset of WizardLM (Xu et al., 2023) for chat, a 100k subset of MetaMathQA (Yu et al., 2024) for math, and a 100k subset of Code-Feedback (Zheng et al., 2024) for code. Before the fine-

| Method | Sparsity | MT-Bench | GSM8k (0-shot) | Human-eval (Pass@5) |
|---|---|---|---|---|
| LoRA | None | 7.58 | 80.21 | 79.4 |
| SparseGPT & LoRA | None | 6.11 | 67.93 | 51.8 |
| SparseGPT & SP-LoRA | 2:4 | 5.91 | 67.85 | 49.4 |

Table 5: Evaluation results of pruned Llama-3.1-8B-instruct model that continually pre-trained on the FineWeb-edu-5B and fine-tuned on Meta-Math, CodeFeedback, and WizardLM.

tuning, we first continually pre-train the model on a subset of FineWeb-edu dataset (Penedo et al., 2024) with 5B tokens and Stanford Alpaca dataset. Then, we fine-tune the model on three datasets WizardLM, MetaMathQA, and Code-Feedback, respectively. Finally, we tested the model's performance in each domain on the benchmarks MT-Bench (Zheng et al., 2023), GSM8K (Cobbe et al., 2021), and Human-eval (Chen et al., 2021) respectively. During the training, the rank of adapters is set to 128, the batch size is set to 256k tokens for the FineWeb-edu dataset and 32 samples for domain-specific data and the Stanford Alpaca dataset, and the learning rate is set to $2 \times 10^{-4}$.

All the training and testing processes are conducted on Nvidia A800-80G GPU and Nvidia A6000-48G GPU.

**Baselines** We evaluated models trained using SP-LoRA against both the original dense models and those pruned by SparseGPT and Wanda. We also compared SP-LoRA with LoRA, a well-known parameter-efficient tuning method for LLMs, and SPP, an existing sparsity-preserving tuning method for sparse LLMs. Beyond evaluating model performance, we also measured each approach's training time and memory overhead.

## 4.1 Main Results

Table 1 and Table 3 illustrate the zero-shot performance of the Llama-2-7B, Llama-2-13B, and Llama-3-8B models, along with their respective versions that were pruned and fine-tuned using the SlimPajama-0.5B and Stanford Alpaca datasets.

The experimental outcomes indicate that SP-LoRA enhances the performance of sparse models, demonstrating an improvement ranging from 2% to 9% over sparse models derived through post-training pruning techniques. Furthermore, SP-LoRA performs similarly to established methodologies such as LoRA and SPP. Notably, while LoRA effectively improves the performance of pruned LLMs, this approach diminishes practical usability due to the resultant dense model. Conversely, SPP relies on tensor parallelism (Shoeybi et al., 2020) to mitigate the high memory footprint associated with sparse LLMs training, limiting its applicability in resource-constrained environments. At the same time, it may also introduce additional communication overheads when considering scenarios of parallel training through multiple GPUs. A detailed comparative analysis between SPP and SP-LoRA is provided in Appendix A. It is important to acknowledge that our training involved a constrained dataset; hence, augmenting the volume of training data would likely yield further enhancements in model performance, as evidenced in Table 4.

Tables 1 and 2 indicate that we utilized the SlimPajama (pre-training data) and Stanford Alpaca (instruction data) datasets for fine-tuning, observing that the resulting models exhibit comparable performance. However, the perplexity scores on the wikitext2 dataset, as shown in Table 2, reveal a significant discrepancy. Fine-tuning with the pre-training data results in lower perplexity compared to fine-tuning with the instruction data. This suggests that instruction fine-tuning data may be more effective in enhancing performance on downstream tasks than pre-training data. While existing methods, such as SPP, evaluate sparse models trained on instruction fine-tuned datasets against the base model, our findings suggest that utilizing pre-trained data for comparisons might provide a more equitable assessment.

To evaluate the domain adaptation capabilities of SP-LoRA, we conducted experiments using the Llama-3.1-8B-instruct model. Initially, the model was pruned using SparseGPT. Subsequently, to restore the model's performance, we employed SP-LoRA for fine-tuning alongside the FineWeb-edu-5B and Alpaca datasets. The evaluation results of the fine-tuned sparse model are presented in

Table 4. Furthermore, we fine-tuned both the dense and sparse models using LoRA and SP-LoRA on the WizardLM, MetaMathQA, and Codefeedback datasets, respectively. The models were then evaluated on the MT-bench, GSM-8k, and Huam-Eval benchmarks, as summarized in Table 5. Our results indicate that the fine-tuned sparse model achieves approximately 78% of the performance level of the dense model on chat tasks, 85% of the performance level on mathematical tasks, and 65% of the performance level on coding tasks. At the same time, SP-LoRA has a competitive performance compared to LoRA in fine-tuning sparse model. In terms of code-related task Human-Eval, SP-LoRA exhibits poorer performance. A potential reason for this could be the lack of code data during continuous pre-training. We posit that the performance of the sparse model could be further enhanced by supplementing additional code data.

## 4.2 Time and Memory Overhead

In addition to model performance, we also evaluate the time and memory overhead of fine-tuning the sparse LLM using different methods, including LoRA, SP-LoRA with our proposed memory optimization (SP-LoRA), SP-LoRA with gradient checkpointing optimization (SP-LoRA(GC)), SP-LoRA with no optimization (SP-LoRA(NO)), SPP with gradient checkpointing optimization (SPP(GC)), and SPP with no optimization (SPP(NO)). The implementation details of these methods are presented in Appendix B. We performed our experiments on a single Nvidia A6000 GPU with the batch size set to 1 and the sequence length set to 2048. The experimental results are shown in Figure 1. It can be seen that SP-LoRA outperforms SPP(GC) and SPP(NO) in terms of speed and memory overhead, where SPP(NO) leads to out-of-memory error, and gradient checkpointing significantly reduces SPP(GC)'s training speed. Also, SP-LoRA is faster and uses less memory than SP-LoRA(GC), while significantly reducing memory usage compared to the SP-LoRA(NO). Finally, compared to LoRA, SP-LoRA has similar time and memory overheads. All these results demonstrate the effectiveness of our approach.

## 5 Conclusion and Future Works

In this paper, we introduce the SP-LoRA method, which is a parameter-efficient and memory-efficient approach for training sparse models while preserving the sparsity. Our approach addresses the challenges of domain adaptation and performance restoration for sparse LLMs. Specifically, we introduce additional masks in the LoRA framework, thus preserving the sparsity of the LLM during training, and achieve memory efficiency by using a hybrid gradient checkpointing and memory reutilizing approach. Experiments on the Llama family show that SP-LoRA can effectively recover the performance of pruned LLMs and has comparable performance to LoRA on domain migration tasks.

Currently, in the SP-LoRA framework, we only consider static masks, and at the same time, we do not use LoRA variants to further improve the performance of SP-LoRA. Therefore, looking ahead, we will try to use different improved versions of LoRA combined with dynamic mask tuning methods for better performance.

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

## A  COMPARISON BETWEEN SPP AND SP-LORA

SPP (Lu et al., 2024) is also a parameter-efficient and sparsity-preserving fine-tuning methodology. The formulation of SPP can be mathematically described as follows:

$$\tilde{\mathcal{W}}^{(t)} = \tilde{\mathcal{W}} + \tilde{\mathcal{W}} \odot \text{Repeat}_1(\mathcal{A}^{(t)}, \frac{C}{r}) \odot \text{Repeat}_0(\mathcal{B}^{(t)}, R), \tag{8}$$

where $\tilde{\mathcal{W}} \in \mathbb{R}^{R \times C}$ denotes the updated weight matrix, $\mathcal{A} \in \mathbb{R}^{R \times r}$ and $\mathcal{B} \in \mathbb{R}^{1 \times C}$ represent the learnable parameter matrices, and $\text{Repeat}_i(x, n)$ means repeating the tensor $x$ along axis $i$ for $n$ times. The adjustment to the weight matrix, denoted by $\tilde{\mathcal{W}} \odot \text{Repeat}_1(\mathcal{A}^{(t)}, \frac{C}{r}) \odot \text{Repeat}_0(\mathcal{B}^{(t)}, R)$, is formulated as the Hadamard product of these three matrices, thereby maintaining the sparsity structure inherent in the matrices involved. Furthermore, the parameters $\mathcal{A}^{(t)}$ and $\mathcal{B}^{(t)}$ are the only ones subject to training, which significantly reduces the parameters compared to that of $\tilde{\mathcal{W}}$, thus exemplifying the parameter efficiency of this approach.

It is observed that SPP can be conceptualized as a variant of LoRA. To illustrate this perspective, consider partitioning each sequence of $r$ consecutive elements within $\mathcal{B}$ into segments, such that:

$$\mathcal{B} = [\mathcal{B}_1, \mathcal{B}_2, \ldots, \mathcal{B}_{\frac{C}{r}}], \tag{9}$$

where each segment $\mathcal{B}_i$ is a vector of length $r$. Subsequently, we define a block-diagonal matrix $\hat{\mathcal{B}}$ constructed from these segments:

$$\hat{\mathcal{B}} = [\text{diag}(\mathcal{B}_1), \text{diag}(\mathcal{B}_2), \ldots, \text{diag}(\mathcal{B}_{\frac{C}{r}})]. \tag{10}$$

With this definition, the update rule for the weight matrix $\tilde{\mathcal{W}}$ can be rewritten as:

$$\tilde{\mathcal{W}}^{(t)} = \tilde{\mathcal{W}} + \tilde{\mathcal{W}} \odot (\mathcal{A}^{(t)} \times \hat{\mathcal{B}}^{(t)}). \tag{11}$$

Therefore, SPP can be interpreted as a LoRA variant that employs a specialized matrix $\hat{\mathcal{B}}$, augmented with the initial weight matrix $\tilde{\mathcal{W}}$ as a weight term, to achieve its parameter-efficient and sparsity-preserving properties.

Recalling the mathematical form of the SP-LoRA,

$$\tilde{\mathcal{W}}^{(t)} = \tilde{\mathcal{W}} + \mathcal{M} \odot (\mathcal{A}^{(t)} \times \mathcal{B}^{(t)}). \tag{12}$$

The distinctions between SPP and SP-LoRA can be delineated as follows:

- SPP employs a composite weight matrix $\hat{\mathcal{B}}$ formed by stitching together multiple diagonal matrices, whereas SP-LoRA utilizes a standard matrix $\mathcal{B}$ as its weight matrix.
- SPP incorporates the initial weight matrix $\tilde{\mathcal{W}}$ as an additional weight term, while SP-LoRA leverages a mask matrix $\mathcal{M}$ as an additional weight term.

Incorporating the initial weight matrix $\tilde{\mathcal{W}}$ as an additional weight term endows SPP with certain advantages in instruction fine-tuning. However, this approach precludes SPP from benefiting from the proposed memory reuse technique and poses the challenge of high GPU memory overhead. To solve the problem of high GPU memory usage, SPP uses tensor parallelism, where the weight matrices are sliced and stored separately within different GPUs. However, this optimization requires multiple GPUs to implement and thus cannot be applied to low-resource fine-tuning scenarios with only a single GPU. Also, in multi-GPU parallel training scenarios, SPP enforcing the use of tensor parallelism may reduce the training speed due to the increased communication overhead.

Conversely, the proposed method, SP-LoRA, achieves comparable time and memory overheads to those of LoRA through optimized memory usage, while simultaneously maintaining equivalent performance levels as SPP.

# B  IMPLEMENTATION OF SPP AND SP-LoRA VARIANTS

```python
def forward_adapter(x, W, A, B):
    n, m = W.shape
    r = A.shape[1]
    A = torch.repeat_interleave(weight, m // r, dim=1)
    B = torch.repeat_interleave(weight, n, dim=0)
    W_adapter = W * A * B
    return F.linear(x, W_adapter)

def forward_spp(x, W, A, B):
    y1 = F.linear(x, W)
    y2 = forward_adapter(x, W, A, B)
    return y1 + y2
```

Listing 1: Implementation of SPP(NO)

```python
def forward_adapter(x, W, A, B):
    n, m = W.shape
    r = A.shape[1]
    A = torch.repeat_interleave(weight, m // r, dim=1)
    B = torch.repeat_interleave(weight, n, dim=0)
    W_adapter = W * A * B
    return F.linear(x, W_adapter)

def forward_spp(x, W, A, B):
    y1 = F.linear(x, W)
    # gradient checkpointing
    y2 = checkpoint(forward_adapter, x, W, A, B)
    return y1 + y2
```

Listing 2: Implementation of SPP(GC)

```python
def forward_adapter(W, A, B):
    M = (W != 0)
    return W + M * (A @ B)

def forward_sp_lora(x, W, A, B):
    W_new = forward_adapter(W, A, B)
    return F.linear(x, W_new)
```

Listing 3: Implementation of SP-LoRA(NO)

```python
def forward_adapter(W, A, B):
    M = (W != 0)
    return W + M * (A @ B)

def forward_sp_lora(x, W, A, B):
    # gradient checkpointing
    W_new = checkpoint(forward_adapter, W, A, B)
    return F.linear(x, W_new)
```

Listing 4: Implementation of SP-LoRA(GC)

