# OpenReview forum: "SP-LoRA: Sparsity-Preserved Low-Rank Adaptation for Sparse Large Language Model"
_ICLR.cc/2025/Conference — ICLR 2025 Conference Withdrawn Submission_

### Official Review · Reviewer_vHup · 2024-10-25

**Soundness:** 3
**Presentation:** 4
**Contribution:** 2
**Rating:** 5
**Confidence:** 5

**Summary:**

This paper introduces SP-LoRA, which is a memory-efficient sparsity preserved training method. SP-LoRA can recover the accuracy of pruned LLMs with similar memory and time usage of LoRA by proposing a hybrid technique which combines gradient checkpointing and memory reuse.

**Strengths:**

-Memory efficient training is an important topic for LLMs.

-Extensive experiments have been conducted to prove the effectiveness of SP-LoRA

-Paper writing is easy to follow, the formula derivation is clear enough.

**Weaknesses:**

-My major concern is about the research contribution of this paper. Although the techniques such as memory reutilization are effective for memory-efficient sparsity preserved training, they seem to be general solutions to this kind of problem, thus making this work more of an engineering contribution rather than research one. Maybe a more detailed explanation of the challenges of combining all techniques together can make this work better.

-A better baseline is needed to show the Sec 3.2.1 memory complexity. As kernel fusion or in-place computation are widely used in famous frameworks such as FasterTransformer, it seems to be a poor baseline shown in Formulation 3.

-In Figure 1, SPP(GC) has similar memory usage with SP-lora, although slower than SP-lora, but only 2 seconds per batch. As the number of batches needed for fine-tuning is generally small, SPP(GC) seems to be fast enough for fine-tuning. Therefore, the contribution of SP-lora over SOTA method SPP(GC) seems to be limited.

**Questions:**

Most can found in Weakness.

Another suggestion:

-Figure quality (e.g. font size) can be improved, e.g. Figure 1.

---

> ### Author Response · Authors · 2024-11-17
> **Reply to the weakness**
>
> Thank you for your valuable review comments. We acknowledge the weaknesses that you have pointed out and will carefully revise our paper to address these points in future versions of our work.

---

> > ### Comment · Reviewer_vHup · 2024-11-17
> > **Reply to the author's response**
> >
> > Thanks for your response. I am maintaining my score.

---

### Official Review · Reviewer_NSkJ · 2024-11-03

**Soundness:** 1
**Presentation:** 2
**Contribution:** 1
**Rating:** 3
**Confidence:** 4

**Summary:**

This paper introduces SP-LoRA, a method for fine-tuning sparse LLMs while preserving sparsity, addressing memory and efficiency challenges. Unlike existing LoRA method, SP-LoRA introduces mask matrix into low-rank adaptation, allowing sparse models to retain their sparsity during fine-tuning. To address the memory demands of sparse fine-tuning, SP-LoRA combines gradient checkpointing and memory reuse to optimize GPU memory consumption. Experiments demonstrate the approach's effectiveness.

**Strengths:**

1. This paper is well-written, and the author provides a clear analysis of the increased memory overhead introduced by the mask matrix, making it easy to understand.
2. SP-LoRA consumes less memory than SPP baseline when fine-tuning sparse LLMs and also has an advantage in fine-tuning time.
3. After fine-tuning with SP-LoRA, sparse low-rank adaptation can be merged into sparse LLMs, preserving the sparsity of the LLMs.

**Weaknesses:**

Although the SP-LoRA method has the above strengths, I think it includes but is not limited to the following weaknesses:

1. **Lack of novelty**. Sparsity-Preserved Parameter-Efficient Fine-Tuning (SPP) for LLMs has already been proposed, and the Sparsity-Preserved Low-Rank Adaptation (SP-LoRA) introduced here is not novel and lacks groundbreaking contributions.
2. **Limited contribution**. The author proposes using gradient checkpointing and memory reuse techniques to optimize SP-LoRA’s memory overhead and fine-tuning efficiency. These are well-established techniques implemented using PyTorch APIs. The author merely combines existing techniques, making SP-LoRA appear more like an engineering improvement.
3. **Accuracy is worse than or only slightly improved over baselines**. The data in Tables 1-5 shows that SP-LoRA’s accuracy is much lower than that of the LoRA and SPP baselines or only offers a minimal accuracy improvement. Although SP-LoRA has memory and time advantages over the baselines, its disadvantage in accuracy makes me very concerned about the effectiveness of SP-LoRA.
4. **2:4 sparsity is not enough**. The author only conducted experiments with 2:4 semi-structured sparsity, which is insufficient to validate the effectiveness of the SP-LoRA method. For example, the author should provide more experimental data with 2:8 sparsity. Additionally, there is a lack of experimental data with 50% or 60% unstructured sparsity, as commonly set in methods like Wanda and SparseGPT.
5. **Experiments are conducted only on LLaMA**. The authors only conducted experiments on the LLaMA family and lacked verification of the effectiveness of SP-LoRA method on models with non-LLaMA architecture (such as OPT model [1]).

Due to SP-LoRA's lack of novelty, limited contributions, lower accuracy compared to the baselines, and insufficient experimentation, I believe this paper is not yet ready to be accepted. I would be happy to discuss my evaluation with the author and look forward to their response.

[1] Opt: Open pre-trained transformer language models. Arxiv.

**Questions:**

I hope the authors can address the aforementioned weaknesses, and I have the following questions that I would like them to respond to.

1. **Rank size**. For pre-training and instruction data, the author sets the rank to 16, whereas for domain-specific data, the rank is set to 128. The difference is substantial, and I would like to know how the rank size affects the final accuracy. Additionally, does the rank size impact fine-tuning memory and time usage? Could the author provide some ablation studies? Also, what are the rank sizes of the compared SPP and LoRA methods?
2. **Dataset setup**. Why is the sparse LLMs first continually pre-trained on the FineWeb-edu and Stanford Alpaca datasets and then fine-tuned on WizardLM, MetaMathQA, and Code-Feedback, respectively? Wouldn’t it be feasible to fine-tune the sparse LLMs directly on the domain-specific datasets? Additionally, why did the author choose FineWeb-edu and Stanford Alpaca for pre-training rather than the SlimPajama dataset?
3. **Why is the accuracy better than LoRA in some cases**? With 2:4 sparsity, the fine-tuning parameter of SP-LoRA is only 50% of LoRA. Given the same fine-tuning data and experimental setup, the accuracy of SP-LoRA should theoretically be lower than LoRA. Why is the accuracy better than LoRA in some cases?
4. **Fine-tuning time**. How much time did the author spend on fine-tuning for pre-training, instruction, and domain-specific data, respectively?
5. **Larger models**. The SPP paper presents experimental results for 30B and 70B models. Does SP-LoRA support 30B and 70B LLMs?

---

> ### Author Response · Authors · 2024-11-17
> **Reply to the questions**
>
> Thank you very much for your valuable feedback. Below, I respond to each of your questions
>
> Q1: To maximize the performance recovery of the pruned model, we plan to use more data for further training, which is why we opted for a larger rank size. In all our experiments, we maintained a consistent rank size across different methods. Your suggestion for ablation studies is greatly appreciated, and we intend to include these in future versions of our work.
>
> Q2: To enhance the model's performance through continuous training. Meanwhile, FineWeb-edu has higher quality than SlimPajama, so in the final experiment, we replaced SlimPajama with FineWeb-edu.
>
> Q3: While we are unable to provide a theoretical explanation at this stage, we note that similar experimental observations have been reported in related work [1].
>
> Q4: The pre-training phase typically requires between one to two days, whereas other tasks generally take only a few hours.
>
> Q5: Although SP-LoRA is applicable to models with 30 billion and 70 billion parameters, we currently lack the computational resources necessary to conduct such large-scale training.
>
> Finally, we would like to clarify one more point that our approach cannot be implemented through the gradient checkpoint API provided by PyTorch.
>
> [1] Muñoz J P, Yuan J, Jain N. SQFT: Low-cost Model Adaptation in Low-precision Sparse Foundation Models[J]. arXiv preprint arXiv:2410.03750, 2024.

---

> > ### Comment · Reviewer_NSkJ · 2024-11-17
> > **Reply to the author's response**
> >
> > I have read the author's response; however, they did not address all of my weaknesses and questions and failed to provide additional ablation studies, such as experiments with 50% or 60% sparsity, rank ablation experiments, and experiments on different datasets. Additionally, I have reviewed the comments from other reviewers and agree with their concerns regarding the research contributions and the completeness of the experiments in this paper. I find that the authors have not addressed the issues raised by other reviewers at this stage. Therefore, I will maintain my score and am inclined to reject this paper.

---

> > > ### Author Response · Authors · 2024-11-17
> > > **Thanks for your feedback**
> > >
> > > Thank you very much for your timely feedback. Given the current review score and the substantial amount of additional content required, we have decided to withdraw the paper following the rebuttal stage. As such, we have not included extra experimental results in our response. Even though we intend to withdraw the manuscript, we are open to further discussion and would welcome any additional comments or questions you may have at this stage.

---

### Official Review · Reviewer_TpEh · 2024-11-03

**Soundness:** 2
**Presentation:** 3
**Contribution:** 2
**Rating:** 5
**Confidence:** 3

**Summary:**

Pruning methods present a promising angle for making LLMs more computationally efficient. This work focused on fine-tuning already sparsified models using a masked LoRA-like low-rank adapter. To reduce the memory requirements caused by the additional masking step, the authors implement gradient checkpointing and in-place reuse of the constant weight matrix across time steps. The experimental results demonstrate similar time and memory usage as LoRA while improving over sparse models derived via post-training pruning.

**Strengths:**

The work tackles an important open problem of developing fine-tuning methods for sparse LLMs by considering a straightforward masking extension of the LoRA low-rank adaption technique.

The paper does not only evaluate the algorithm under abstract performance metrics but is also concerned with concrete, memory- and time-efficient hardware implementation on GPUs. The detailed memory analysis of Section 3 is helpful for considering the challenges and trade-offs of sparse LLM training.

The experimental evaluations include a comparison against original dense models and alternative sparsification methods across several models, demonstrating the method's viability for both zero-shot and domain-specific tasks.

**Weaknesses:**

While the approach that the authors propose is straightforward, it comes with a high memory overhead compared to the baselines. This is not necessarily an issue in itself since much of the paper is devoted to addressing this memory problem. However, since it is the SP-LoRA method that introduces the memory overhead in the first place, I would expect SP-LoRA not just to match but outperform the baselines in some respect. In other words, the memory cost that the masking introduces should be worth the trouble on some other metric. As far as I can tell, SP-LoRA performs similarly to LoRA and SPP. In particular, the memory cost after optimization appears to be the same (Figure 1). Consider including an experiment that shows the inference memory usage of the resulting sparse SP-LoRA network compared to the baselines. Given that LoRA does not preserve sparsity, SP-LoRA may show significant advantages at inference time that could justify the memory overhead during fine-tuning. Alternatively, consider exploring scenarios where preserving sparsity might be particularly beneficial, like when scaling to larger model sizes. Evaluating the memory usage for growing model sizes could reveal a favorable trade-off that could justify using SP-LoRA over the baselines as model size grows.

Additionally, Figure 1 suggests that SP-LoRA reduces time usage compared to SPP. Consider adding a more detailed analysis of the time usage differences across different model sizes or sparsity levels. This would help clarify whether the time advantage is consistent and significant. If there is a general advantage independent of implementation details, it would provide another reason to use SP-LoRA over SPP. Furthermore, Appendix A contains a valuable analysis of the specific differences between LoRA and SPP. It would be worth exploring these differences in an ablation study that compares the usage of a composite matrix vs a standard matrix and the usage of mask vs additional weight term. The study could evaluate training time, memory usage, and task performance across different sparsity levels and thus provide a clearer picture of the trade-offs involved in sparse training when using these two methods.

Suggestions to improve the clarity of the paper:
- The introduction is written with the assumption that the reader is familiar with LoRA's definitions and terminology, which, however, are not introduced until Section 3.1. To make the introduction more accessible to a broader audience, consider giving readers a brief idea of LoRA and the extension this work proposes.
- L73-L78+L82-L87: I appreciate the effort to formulate a concise overview of the method in the introduction, but this early on in the paper without any preliminaries in place, I found it hard to follow and diminished the clarity of the introduction. Consider focusing on the key idea behind your work in the introduction, deferring the mathematical details to Section 3.
- L464 To aid the reader, could you please reference which experimental outcomes specifically show this improvement
- For the Tables:
	- Consider highlighting the best average scores in bold.
	- There is a typo 'Mehtod' in Table 1 and 3.
    - Instead of denoting the sparsity type 2:4 that is the same for all methods, consider listing the weight sparsity levels and non-zero weight count.

**Questions:**

L493 suggests that a reason for SP-LoRA's poorer code task performance could be the lack of code data during pre-training. I am not sure why this would not also affect the LoRA performance. Is there something specific about SP-LoRA that gives you a reason to expect a poorer code task performance?

Can the proposed memory recomputation and reuse strategies also be applied to LoRA, and if so, would you expect further performance improvements for the LoRA baseline?

---

> ### Author Response · Authors · 2024-11-17
> **Reply to the questions**
>
> Thank you for your valuable feedback. Below, I respond to each of your questions.
>
> Q1: Sorry for the potential misunderstanding. Our intention was to explain why both LoRA and SP-LoRA show a significant performance gap compared to the baseline, rather than why SP-LoRA performs lower than LoRA.
>
> Q2: For very long sequences, our method, which involves merging the adapters with the weight matrices during computation, can reduce GPU memory usage compared to the existing LoRA method. This is particularly beneficial for handling longer sequences where memory efficiency is crucial.

---

> > ### Comment · Reviewer_TpEh · 2024-11-17
> >
> > Thanks for your response. I recommend clarifying these points in future revisions of the manuscript. I am maintaining my score.

---

### Official Review · Reviewer_kGAz · 2024-11-04

**Soundness:** 3
**Presentation:** 3
**Contribution:** 3
**Rating:** 5
**Confidence:** 3

**Summary:**

This manuscript proposes a fine-tuning method by adopting LoRA while preserving sparsity by introducing a mask. This mask requires extra memory, to address this issue, the authors utilized gradient checkpoints with memory reuse. This fine-tuning method can be applied on top of sparse LLMs (pruned by Wanda/ SparseGPT) and can further optimize the LLMs to restore the performance. The authors provided the algorithms for computation of mask and gradients in the forward and backward pass. The authors showed the effectiveness of the proposed method by zero-shot evaluation on seven datasets and by domain-specific tasks in three datasets. The performance of these fine-tuned models are better than the baselines and similar to its' competitors such as wanda and sparseGPT.

**Strengths:**

1. this work addresses an important gap in the field by enabling efficient fine-tuning of sparse Large Language Models (LLMs) while preserving their sparsity.
2. The proposed hybrid technique combining gradient checkpointing and memory reuse effectively reduces GPU memory usage during fine-tuning, which is much comparable to standard LoRA.
3. The method is tested on three smaller LLMs with different sparsity patterns and ratios, where it showed its versatility and effectiveness.
4. this work works both on zero-shot performance and domain-specific fine-tuning. also, this work has a backward path algorithm, so training/ fine-tuning is possible.
5. The manuscript provides detailed explanations of the method with mathematical background and visual representations of the workflow.
6. This method works with smaller models, as seen from the results with zero-shot tasks and domain-specific tasks.

**Weaknesses:**

1. The manuscript does not discuss any specific kernel implementations or hardware-level optimizations for SP-LoRA, which could limit its practical efficiency on specialized hardware.
2. The evaluations appear to be primarily conducted on smaller to medium-sized LLMs. It's unclear how SP-LoRA would scale to very large language models with hundreds of billions of parameters.
3. The method is mainly tested on models pruned by Wanda or SparseGPT, which may not cover all possible sparsity patterns. It's uncertain how SP-LoRA would perform with other sparsity types or pruning methods.
4. The manuscript could benefit from more extensive comparisons with a wider range of models and other parameter-efficient fine-tuning techniques beyond LoRA and SPP.
5. While the manuscript mentions evaluations on domain-specific tasks like math and code, the results presented for these tasks may not be comprehensive enough to fully demonstrate SP-LoRA's capabilities across various domains.
6. Although memory usage is optimized, the method may still introduce some computational overhead compared to standard LoRA, which could impact training time.

**Questions:**

1. How does the effectiveness of SP-LoRA vary with different initial pruning techniques? Is there an optimal approach for pruning before applying SP-LoRA? For instance, how does it compare with structural pruning methods like those proposed in LLM-Pruner?
2. What impact does SP-LoRA have on inference speed compared to dense models and other sparse fine-tuning methods? How does it compare to quantization methods like SpQR in terms of compression and speed?
3. How well does SP-LoRA scale with very large language models (for example, those with hundreds of billions of parameters)? Have you tested it on models larger than the ones mentioned in the paper?
4. Can the principles behind SP-LoRA be applied to other types of neural networks beyond language models? For instance, could it be adapted for multi-modal models or vision models?
5. How does the choice of rank in SP-LoRA influence the balance between performance and efficiency? Does it follow similar patterns to other LoRA-based methods like LoftQ?
6. How does SP-LoRA compare to other sparsity-aware fine-tuning methods in terms of convergence speed and run-time? For example, how does it compare to Sparse Low-rank Adaptation?
7. Could SP-LoRA be combined with quantization techniques like those in LoftQ or SpQR for even greater compression? What challenges might arise from such a combination?
8. How does SP-LoRA handle the trade-off between sparsity and model quality? Is there a point at which increasing sparsity significantly degrades performance?

---

> ### Author Response · Authors · 2024-11-17
> **Reply to the questions**
>
> Thank you for your valuable feedback. Below, I respond to each of your questions.
>
> Q1: The primary goal of SP-LoRA is to fine-tune a sparse neural network while preserving its sparsity. This means that the specific sparse neural network we end up with depends on the pruning algorithm used. Based on our experiments, we found that the sparse network with the best initial performance typically yields the best results post-fine-tuning. Additionally, LLM-Pruner employs a structured pruning approach, meaning the resulting model remains dense. Therefore, it does not align with our method. Furthermore, at equivalent compression rates, unstructured pruning tends to outperform structured pruning in terms of performance.
>
> Q2: SP-LoRA is designed as a fine-tuning technique. Similar to LoRA, the adapter can be merged with the weight matrix when deployed, ensuring no extra computational cost during inference. The inference speed largely hinges on how efficiently sparse matrix multiplications are implemented across various hardware platforms, an aspect that falls outside the scope of our current study.
>
> Q3: Unfortunately, due to constraints in computational resources, we were unable to conduct experiments with larger models.
>
> Q4: Indeed, as long as the methods utilize a sparse matrix, they are compatible with SP-LoRA.
>
> Q5: For determining SP-LoRA’s rank, we adhered to the established practices outlined in previous studies [1, 2].
>
> Q6: Like SPP, our core objective is to maintain the sparsity of the weights throughout the fine-tuning process. This sets our approach apart from other methods, including sparse low-rank adaptation.
>
> Q7: We plan to explore integrating SP-LoRA with quantization-aware training techniques in future research.
>
> Q8: In the context of unstructured pruning, we observed that sparsity levels exceeding 60% to 70% lead to a significant decline in performance.
>
> [1] Hu E J, Shen Y, Wallis P, et al. Lora: Low-rank adaptation of large language models[J]. arXiv preprint arXiv:2106.09685, 2021.
> [2] Zhao J, Zhang Z, Chen B, et al. Galore: Memory-efficient llm training by gradient low-rank projection[J]. arXiv preprint arXiv:2403.03507, 2024.

---

### Note · Authors · 2024-12-01

I have read and agree with the venue's withdrawal policy on behalf of myself and my co-authors.